# Priorities in the Interdisciplinary Approach of Specific Learning Disorders (SLD) in Children with Type I Diabetes Mellitus (T1DM). From Theory to Practice

**DOI:** 10.3390/brainsci11010004

**Published:** 2020-12-23

**Authors:** Paraskevi Tatsiopoulou, Georgia-Nektaria Porfyri, Eleni Bonti, Ioannis Diakogiannis

**Affiliations:** 1st Department of Psychiatry, School of Medicine, Faculty of Health Sciences, Aristotle University of Thessaloniki, “Papageorgiou” General Hospital, Ring Road Thessaloniki, N. Efkarpia, 54603 Thessaloniki, Greece; geoporfyri@hotmail.fr (G.-N.P.); elina.bonti@gmail.com (E.B.); idiakogiannis@auth.gr (I.D.)

**Keywords:** specific learning disorders (SLD), type I diabetes mellitus (T1DM), narrative review, interdisciplinary approach, children and adolescents

## Abstract

Background: A considerable endeavor had taken place in order to understand the associated challenges for children and adolescents with Specific Learning Disorder (SLD) and Type 1 Diabetes Mellitus (T1DM) but also in order to describe the necessary skills and approaches that the care givers have to develop to assist both children and parents. (1) Aim: The aim of this review is twofold. Firstly, to highlight the T1DM’s potential impact on psychological well-being, on cognitive functioning and on school performance in children and adolescents who confront SLD. Secondly, to discuss the necessity of a multidiscipline approach of poor school performance in students with SLD and T1DM, presenting the serious contribution of care providers: (a) parents/carers in the family setting, (b) teachers and psychologists in the school setting and (c) health specialists (pediatricians, nutricians, nurses, child psychiatrists and psychologists) in the medical setting. (2) Methods: In this narrative literature review of 12 selected articles, each one studies a special aspect of approach, during the diagnosis and the treatment of individuals with T1DM and SLD. The review concerns the arising problems and difficulties in the adherence to diagnosis, the management of insulin, the mental and physical wellbeing, the school performance, the cognitive functioning and learning difficulties of patients. We tried to synthesize an interdisciplinary approach that involves collaboration between family, school and medical frame; facilitating children’s and adolescents’ difficulties management, as well as parent and teacher involvement during the intervention implementation. (3) Results: The main issues of concern were examined through the available literature, as different factors had to be re-examined in the previous studies, regarding the potential impact of T1DM in cognitive and psychological functioning, as well as the effects of the intervention/approach/treatment of children and adolescents with SLD and T1DM. (4) Conclusions: Although T1DM diagnosis and demanding treatment are a heavy burden for children and their families, T1DM may or may not be associated with a variety of academic and psychological outcomes. Despite the variability of the reviewed research design quality, it was clearly defined that the impact of T1DM is not uniform across educational and mental variables. Strengthening the children’s physical, psychological and social wellbeing is an especially important factor, as it facilitates the insulin’s management as well as the learning difficulties. This is possible by supporting the parental and teacher involvement in the intervention process. This review highlights the need to reduce the distance between theory/research and practice, in some of the proposed areas in this field of knowledge.

## 1. Introduction

School performance, especially in children with specific learning disorders (SLD), is adversely affected by the coexistence of a chronic disease, such as Type 1 Diabetes Mellitus (T1DM) (previously called juvenile diabetes or insulin-dependent diabetes). Being diagnosed with a chronic illness can be overwhelming; especially at the start it can be stressful for the child and its family. We are conscious that living with T1DM can be really challenging for children, adolescents, and their families, regarding the complexity of the treatment, the involvement of the adult care givers and the needed support from the school setting. Children with T1DM and SLD confront major challenges, managing both insulin and learning difficulties within the school setting.

Data from studies reviewed in this research, show that students with learning difficulties have higher rates of diabetes compared to the general population. The report of Public Health England, NHS Digital in 2016, supports the occurrence of higher rates of both diabetes types, in all age groups, in the population with SLD compared to the general population, and with early onset, recorded at a younger age [1,2]. In addition, a greater risk of autoimmune manifestations-including T1DM-that is recorded in Down’s syndrome [3], could explain the T1DM leading rates among children and adolescents with SLD [2]. Furthermore, SLD is also associated with higher rates of developing Type 2 Diabetes Mellitus (T2DM) later in adulthood [4,5,6], higher risk of obesity, due to a lifestyle with low level of exercise and high-fat diets, as well as higher levels of prescribed antipsychotic medication [2,7].

Initially, since in literature we find only a few facts about the coexistence/comorbidity of SLD and T1DM, it is important to clarify some points about prevalence and phenomenology that characterize them.

### 1.1. Specific Learning Disorders (SLD)

Specific learning disorder (SLD) is also referred as learning disorder/disability, representing a neurodevelopmental [8] and neurobiological [9] disorder, that usually begins during the early school-age, and possibly not recognized until adolescence and even adulthood [8,10,11,12]. According to the diagnostic criteria of DSM-5, SLD is characterized by three types of continuous difficulties in the ability of learning, concerning one out of three fundamental domains of reading, writing and math; manifesting as a failure in the development of these skills, in correspondence to the expected for the age grade [8,9,13]. Apart from these three core areas, other disorders, such as memory problems, inattention and difficulties in social interaction, may also contribute fundamentally to failure in school performance, requiring a more specific intervention [13]. If not recognized and managed at an early age, beyond having lower academic achievement, ongoing difficulties may have a negative long-term impact in adult life [8,9,10,14]. Various difficulties, such as low self- esteem, behavioral and social problems, due to school failure, are associated with low academic achievements and dropping out of school in youths; mental distress, unemployment or under- employment later in adult life [8,10,11,15,16,17,18,19]. In numerous studies, SLD reflects different prevalence in relation to age, gender, psychosocial stage of development and environmental features [15]. The comorbidity of SLD with other disorders, is usually associated with more complicated manifestation and severe emotional and behavioral symptoms, that render interdisciplinary intervention crucial [15].

SLD is a multifactorial disorder, caused by inherent or acquired factors affecting brain structure and function [20]. Genetic and family load, developmental factors, cognitive skills, native language, academic degree, environmental factors, such as socioeconomic status, are mentioned in many studies as severe etiological factors [13,15]. In Table 1 various risk factors are defined as predeterminants for SLD, indicating that the prevalence of SLD is increased among children with the mentioned characteristics regarding family history, medical history and socioeconomic status [13,15,21,22,23,24,25,26,27,28,29,30,31,32,33,34,35,36,37,38,39,40,41,42,43,44,45,46,47,48,49,50,51,52,53,54,55,56,57]. A plethora of research studies have indicated that the prevalence of SLD shows considerable cross-national variation [55] and gender variation, with higher rates among boys comparatively to girls [8].

Table 2 shows the differential diagnosis of SLD, as there are conditions with high risk of learning issues in children and adolescents -that may not meet the diagnostic criteria of SLD- and if left untreated, may be confused with SLD [13,17,27,58,59].

The assessment of these conditions includes a series of essential examination, laboratory diagnostic tests, and supplemental appraisal, or more specialized testing and/or referral such as blood lead level, audiological and vision screening tests etc. Qualitative observations and/or the student’s report card can often identify SLD, but to make a formal diagnosis, psychometric testing is needed (Wechsler Intelligence Scale for Children—WISC) [13,27].

The presence of SLD along with the conditions listed above is common [13]. Anxiety disorders [27,62], behavioral disorders [27,37,65,66], depressive disorders [27,29,62], motor delays/disorders [27,65], neurodevelopmental disabilities (such as Attention Deficit Hyperactivity Disorder-ADHD) [27,37,62,65], speech-language delays/disorders [27,29,30,39,43,60], social-emotional problems and substance abuse [27,67] are the most common comorbidities with SLD [27].

Regarding terminology, in the present literature review, the terms: “SLD”, “Specific Learning Disability”, “learning disability”, “learning difficulties” [8,11,12,68]; refer to miscellaneous group of disorders/difficulties revealed through unsuccessful attempts to obtain knowledge, which subsequently could be retrieved and utilized efficiently [13]. The “SLD”, which as a medical term, constitutes a diagnostic terminology [8], usually mentioned as “learning disorder” [8,11,12,68]; and the “Learning disability”, as an academic and legal term [11,12]; are not precisely identical. “Learning difference” represents a commonly accepted term, contributing in the destigmatization of children and adolescents, helping them reveal and communicate to others the difficulties that they face in learning and school performance, without labeling them as “disordered” [12,68]; or “disabled”, in a sense that the term “learning disability” reveals intellectual disability, formerly mentioned as “mental retardation” [13].

### 1.2. Type 1 Diabetes Mellitus (T1DM)

Type 1 Diabetes Mellitus (T1DM) in children, is also mentioned as “insulin-dependent” or “juvenile” Diabetes, usually firstly diagnosed during childhood and adolescence, can appear at any age and is a life-long disease. It accounts for about 5% of all patients with diabetes, while its incidence and its prevalence are increasing in the world [69,70,71,72].

Childhood and adolescence are periods characterized by prompt developmental transitions and major changes occurring in the brain, which maybe more vulnerable to extremes of glycemia [73,74]. T1DM with an early onset in young age may have a negative impact on the development of the central nervous system (CNS), reflected in the decrement of cognitive and psychomotor efficiency, mental flexibility and attention; due to secondary conditions (such as chronic hyperglycemia, microvascular abnormalities etc.) [73]. As various prospective studies emphasize, the decline in cognitive functioning is associated more with early onset in young age and microvascular complications (such as retinopathy, nephropathy, neuropathy), than with severe hypoglycemia; while higher HbA1c levels are indicative for mental and psychomotor malfunctioning [73,75]. In addition studies in preschoolers, with severe hypoglycemic episodes at a younger age of 5–7 years old, recorded declines in spatial cognition and information recall, indicative of the susceptibility of the developing CNS to severe hypoglycemia [73,76,77]. Other studies, support that both hypoglycemia and microvascular abnormalities, are risk factors for cognitive malfunctioning [73,76,78,79,80]. Adults with T1DM compared to non-diabetics, presented significant decrease in psychomotor functioning, without any difference occurring in the skills of learning, recall, problem solving [73,79,80,81]. Although numerous studies reveal the association between T1DM and structural-functional changes in CNS, there is no etiological association with specific decline of cognitive efficiency. Research using neuroimaging techniques, such as structural MRI studies, highlighted the lower findings in gray and white substance in population with T1DM compared to non-diabetic peers; associated with severe hyperglycemia, early onset and longer duration of diabetes [73,82]. The clinical manifestations of the reduced white substance in patients with T1DM, were associated with inattention and lower performance in speed of information processing and executive function [73,83]. Age of onset and duration of diabetes, along with microvascular complications in intraparenchymal cerebral arterioles are associated with structural changes, specifically, with white matter lesions (WML) [73,83,84,85,86]. 

Diagnosis in children can be overwhelming, as symptoms are individualized, occurring differently in each child, especially in the beginning [87,88]; usually including excessive thirst, dehydration, frequent urination, high levels of glucose in the blood and urine, unusual hunger or loss of appetite, fruity breath, tachypnea, nausea, vomiting, abdominal pain, weakness, fatigue, mood changes and irritability, severe diaper rash, yeast infection in girls etc. [87,88]. The American Diabetes Association (ADA) in the Position Statement “Care of Children and Adolescents With Type 1 Diabetes”, published in 2005 [70,89], highlighted the essential differentiations of diabetes with early onset in childhood, from adult diabetes; regarding developmental stage, epidemiology, pathophysiology, as well as care response [70,90,91]. In children and adolescents, the management of diabetes must not be concluded from adult diabetes therapy, but from the awareness of child’s developmental stage and needs, as well as environmental context. Punctual interdisciplinary intervention is a nodal point in preventive care for children and their families [70].

## 2. Aim

Reporting the lack of literature data for children who experience special learning difficulties (SLD) and school failure, while struggling to adjust with diabetes type 1 (T1DM), the aim of this article is twofold. Firstly, to highlight T1DM potential impact on psychological well-being, cognitive functioning and school performance in children and adolescents, as recorded in the literature reviewed. Secondly, to discuss the necessity of a multidiscipline approach to academic failure and learning difficulties that may occur in children with SLD who suffer from T1DM, presenting the contribution of: (a) parents/carers in the family setting, (b) teachers in the school setting and (c) health specialists (pediatrician, child psychiatrist, psychologist and nurse) in the medical setting.

## 3. Material and Methods

The main purpose of the present systematic literature review was to indicate: (1) basic considerations about SLD and Diabetes impact on mental health, functionality and school performance of children and adolescents, and (2) what is it proposed about managing both diabetes and learning difficulties, in order to improve physical and mental wellbeing. In collecting the articles, searching through PubMed and Google Scholar, as well as WoS and Scopus, which are representative of quality research; we took under consideration articles that were focused on the impact of SLD and T1DM on children and adolescents, published in peer-reviewed journals, with an open access. The “SLD and Type I Diabetes comorbidity”, “differentiation between SLD and Type I Diabetes”, and “interdisciplinary approach for SLD and Type I Diabetes” were used as keywords. Additionally, several other keywords such as “SLD”, “Type I Diabetes”, “comorbidity”, “multidisciplinary approach”, “interdisciplinary approach”, and “holistic approach” also were used. A total of 41 studies were found in the first search, but several studies were excluded as they did not report important demographic data such as participants’ type of learning disorder (13 studies) as well as participants’ diabetes type (16 in total studies), while the heterogeneous quality of the reviewed studies must be included as a limitation. Finally, a total of 12 studies exploring T1DM were identified and included; 4 studies exploring T1DM impact on cognitive function, learning and school performance of children and adolescents, 3 studies exploring psychological issues and psychiatric comorbidity and 5 studies regarding school interventions; covering a period from 1995 to 2018. The results are depicted in Table 3, while trough a narrative approach we attempted to merge the information from the studies concerning 8600 participants with T1DM across Europe and USA. These sections were made based on two researchers’ opinion, while a third researcher, supervised this work and verified the method and findings.

## 4. Results

Despite the wealth of data from the significant number of studies reviewed, it is worth noting that children with SLD and T1DM are less frequently researched. Through the narrative review significant issues arose from the data of the recruited research. The themes that were deduced are introduced under distinct titles, highlighting the following three points: 

### 4.1. The Potential Impact of Diagnosis and Management of T1DM in Children’s Mental Health and Adherence to Insulin Therapy

We are conscious that living with T1DM can be really demanding compared to other chronic conditions. A higher risk of mental comorbidities is linked with T1DM in childhood [93,94,100,101] and adolescence [70,102,103]. Regarding the prevalence of psychological distress, behavioral and mood disorders in population with T1DM, several studies confirm that despite the fact that in childhood, may not differ from the general population, in adolescence the frequency is 2–3 times higher in comparison to non-diabetic peers [70,102,103]. Mental comorbidities may increase disease’s load for both children and carers, worsening metabolic control [94,100,104,105], leading in further deterioration of microvascular complications and increasing mortality rates [106,107]. Even though most of the children and adolescent patients cope sufficiently with daily glucose controls and insulin treatment, overcome difficulties and withstand challenges, demonstrating incredible resilience [98], some appear to suffer more, experiencing severe mental issues; usually depression, eating disorders [98,108]. Although depression is associated to a moderate degree with maladaptation to treatment of T1DM in children [93], adolescents with depression fail to maintain a sufficient metabolic control, facing a greater risk of exposure to short and long-term complications [98,108,109]. Some studies revealed as predictive factor for psychiatric comorbidities, high HbA1c levels in the early phase of the T1DM onset [94,100]; and highlighted an important clinical problem, estimating that there is a high risk of developing a mental disorder 15–20 years since the T1DM onset (reaching 30% [8]) [94,100]. Psychological well-being is associated with competent metabolic control and a supportive and psychologically healthy environment, that will react timely when depression’s symptoms are identified or suspected in a child or an adolescent diagnosed with T1DM, as a responsible adult needs to secure safe diabetes’s management and require help from a mental health professional [98]. 

### 4.2. The Potential Impact of T1DM on Cognitive Learning Function and Its Relation to Academic Deficits

Children and adolescent diabetic patients must cope with a demanding and complicated daily routine, concerning blood glucose (BG) levels control, insulin injections, diet and exercise; while continuing to live a “normal life” as their peers do. However, T1DM as a chronic disease along with the stressful BG control and psychosocial effects, is associated with a huge negative impact on school performance [106], inattention and lower spelling performance. The later, was related to greater hyperglycemia exposure [79,80]. For children with SLD and T1DM, daily routine, learning skills and academic attendance are burdened with low cognitive efficiency, due to the coexistence of these two conditions [92]. 

Numerous studies suggest that although students with T1DM have an average performance on tests of general intelligence, they may demonstrate mild difficulties in cognitive skills, especially in reading [92]. In addition, there is no clear evidence regarding the impact of nearly undetectable neuropsychological deficiency that may occur in children with T1DM, gradually, on their learning skills [92]. Lamentably, there is no substantive data, as the number of studies of learning difficulties in children with T1DM is limited, concerning small and selective samples, using cross-sectional designs, inconsistent control groups and resulting contradictory conclusions [92]. Regarding the performance of this population on specific neuropsychological tests, opposing results are reported, as some studies showed deficits in verbal intelligence [92,110], memory [92,111,112], motor and visuospatial abilities [92,113,114,115]; whereas others identified deficits in abstract/visual reasoning [92,116], attention [92,111,117], work rate and processing speed [92,114]. The T1DM onset before the age of 7 years old [92,111,112,113,118,119] and school absences [92,115,120], are highlighted as risk factors to neuropsychological deficits. The documented subtle neurocognitive impairments among children with T1DMat several ages may not provide assessable detriment in school performance, even gradually [92], in accordance with studies concluding that severe cognitive impairment with a long-term impact in children with T1DM cannot be associated with the effects of diabetes, with the exception of the attribution to hypoglycemic seizures [92,110,111,112,113,114,115,116,117,118,119,120,121].

Despite these findings, monitoring and preventive treatment of hypoglycemia, seizures or coma are essential to secure learning abilities [92].

### 4.3. Challenges Related to Diabetes Management for Children and Parents

The successful management of T1DM differs significantly among other chronic diseases in children and adolescents, as it requires along with a high complexity intervention and family involvement; also a supportive school environment [96,97,122].There’s no cure for T1DM and although advances in BG monitoring and insulin delivery have improved patients’ quality of life, constant management and ongoing targets and tests can be overwhelming and stressful for both parents and child, as they must learn how to give injections, count carbohydrates and monitor BG.

Diabetes 1 diagnosis constitutes a major crisis for both children and their parents [98]. They experience grief, as they have to confront the life-long nature of the disease and the undercurrent fear of the potential complications [98,123,124,125]. Initially, in the early period, when it is first diagnosed, the young patients usually express sadness, anxiety, irritability, despondency, and negativism in taking insulin or attending school [96,97,98,117,126]. Patients’ parents usually mention that they share with their children feelings of despair, anxiety, along with guilt and worries about the uncertain future [96,97,98,117,126]. These are regular responses that usually occur the first year after diagnosis [96,97,98,117,126]. However, children with underlying maladjustment, may develop in the future, adherence difficulties, psychosocial problems or/and difficulties in metabolic control [98,126,127], that tend to pick in adolescence [70,94,96,97,98,128]. This could be attributed to the developmental changes that take place during “normal” puberty, such as physiological changes and insulin resistance [70,92,98].

## 5. Discussion

The reviewed literature revealed the challenge of patient adherence in managing T1DM but also highlighted the disease’s potential impact on the psychological well-being and school performing, something that is underlined and through the clinical work with this child population. In clinical practice, regarding shaping a diagnostic and therapeutic frame for children with learning difficulties, a key question concerns the effect that the underlying metabolic disorder (due to T1DM) has on the child’s cognitive function and to what extent, the clinical picture of learning disabilities is shaped by the developmental disorder regarding SLD. Research shows that T1DM affects the students’ cognitive abilities and behavior. Young patients that have experienced hypoglycemic episodes demonstrated lower cognitive efficiency with a negative impact in school setting, both in social and academic performance [96,129]. Some behavioral changes in the classroom, due to uncontrolled diabetes may be wrongfully associated with other non- medical situations, such as irritability and fatigue in hyperglycemia [96,130] caused by T1DM, can be seen as a behavioral problem, while the affected ability to focus (due to fluctuation in BG), can be mistaken for ADHD or SLD [96,97].

In addition, in terms of approach (both diagnostic and therapeutic), it is a challenge the fact of indistinguishable roles, between the different specialties involved in the diagnostic and therapeutic process. How is the interdisciplinary approach shaped in practice? What is the role of each expert? How do the Pediatrician, the Child Psychiatrist, the psychologist and the Special Educator work constructively together (a) with each other, (b) with the child and (c) with the family? As this narrative review of the literature reveals, the main require is to form an interdisciplinary intervention that assimilates two major areas: (1) specific learning programs that respond to individual needs, (2) an environment (family, school) that will support changes; a fundamental system that will function concurrently and that will ensure that the needs of both sides (SLD and T1DM) were met.

The suggested recommendations regarding care of children and adolescents with T1DM are mainly based on clinical experience, consensus view and data from cohort studies and not from large clinical trials [70]. A high burden of responsibility is placed on caregivers regarding monitoring of diet, physical exercise, BG levels and insulin administration. Unfortunately, this is needed on the most vulnerable period of the development of the patients, in which the child’s behavior may be unforeseeable, the physiological control may not be easy, and the parenting stress may be excessive [131]. Regardless of the distinct management needs of diabetic children and adolescents, hardly ever are offered clinical services and specific education, while the data of research concerning essential behavioral and psychological issues are restrict both in quantity and range [131].

According to the findings of this review, main priorities in shaping a therapeutic frame, considered to be the following:

### 5.1. Early Intervention

Early intervention is the key for people suffering from both learning disorders and chronic disease. Some studies have shown that in early period of first diagnosed diabetes, children and caregivers adjust to the new challenges of daily control and therapy, gradually, so that subsequently they will have a psychosocial status similar to the one of their non-diabetic peers. However, by the second year after diagnosis, they are more likely to manifest maladjustment and/or depression two times more often than their peers [95,96], rendering this period of the first two years crucial for the subsequent course and outcome prognosis [95,96]. Early identification of these severe manifestations and support of children’s and parents’ knowledge about their condition, could provide a more efficient intervention, protecting children’s learning skills and self-esteem [12,14,68,96]. On the contrary uncontrolled T1DM has an ongoing negative impact on mental health and academic functioning [96].

### 5.2. Patient Knowledge and Consent

Diabetes knowledge is the first step for those suffering, as it will help them understand their condition, and make vital decisions concerning their treatment; leading to a more sufficient management [96,132]. Implicating children and adolescents in the consent procedures in the initial stage of the therapy is crucial for their cognitive functioning. Usually that happens at the age of 12 years old, or when their developmental stage may allow their active involvement [70]. These precautions are crucial for the planning of the treatment frame.

### 5.3. A twofold Therapeutic Frame

The conduct of a twofold therapeutic frame aims to confront the challenges of T1DM insulin management as well as SLD, within the family, school and medical setting. The reviewed research has revealed the potential positive effect of the cumulative support, from different contexts that are part of young patients’ life, as children and adolescents depend on social support systems, such as family and care providers [96,133]. Both families and related social networks must be involved in the psychosocial assessment and therapeutic intervention, supporting youths’ eventually transition to independent self-management of T1DM treatment as adults [70,133,134,135,136].

#### 5.3.1. Medical Setting

Various studies have shown that since first diagnosis, the support of both children and parents is vital [101], due to the high rates of physical and mental complications that may occur from the onset of the disease [99]. The treatment incorporates an interdisciplinary approach, regarding three domains, which are: (1) the information that must be given to patients and parents, so that they have a clear understanding of the social and psychological consequences of T1DM [137]; (2) the psychological group therapy and individual therapy [99]; and (3) the management of diet, physical exercise and lifestyle [70]. Support and education about T1DM self-management must be assured by registered diabetes’ educators, dietitians, exercise specialists, pediatric endocrinologists, nurses, trained mental health professionals -such as child and adolescent psychiatrists (CAP) and psychologists- experienced in childhood diabetes. Mental health specialists by distinguishing between depression, anxiety and diabetes distress, may provide apt therapeutic intervention [70]. Lifestyle management may include youths’ and parents’ training in mitigation methods for both hypo as well as hyperglycemic events [70]. Therapeutic intervention begins when T1DM is firstly diagnosed with an extensive initial training that is updated quarterly by diabetes educator and endocrinologist and annually by dietitian, ensuring the continuing training all through childhood and adolescence [70]. The health professionals help children/adolescents and their parents communicate their concerns, supporting collaboration between them through conflict resolution, they emphasize mutual respect and they actively encourage shared decision-making. In a focused-problem solving intervention, a good starting point is addressing the young patients’ major concern, by investigating the hardest part about living with T1DM at the time of the assessment [137].

In Table 4, we provide recommendations, regarding, how care providers and health professionals may support early adjustment to T1DM treatment [70,98].

Table 5 illustrates the T1DM demands across childhood and the way that the development affects its management; it is an updated form of the original version of the ADA’s Position Statement guidelines for T1DM care among children and adolescents [70,89,131,138]. The management plan of childhood-onset T1DM depends upon the patient’s age, his cognitive ability and mental competence, regarding the risk of hypoglycemia and the development of self- management skills [70,89,138]; aiming to prevent T1DM effects on normal development through childhood and adolescence [98]. 

#### 5.3.2. Group Medical Setting

Through groups and parent-to-parent guidance, mental health professionals provide support, discussing challenges that patients and parents confront in daily routine regarding the T1DM management [115]. At every meeting, expectations and goals are reviewed so that they are coincident with the developmental stage and age of the youths and also support is provided to children and adolescents in order to develop skills that will help them respond sufficiently to the glycemic control [108,139,140].

#### 5.3.3. Family Setting

Through the early years of childhood, parents have a larger responsibility and supervision of T1DM management, while during adolescence, even if the primary caregiver remains a major part of T1DM management, a gentle transition occurs to other caregivers, such as school personnel [70]. A diabetes care with a family- centered approach ensures that psychosocial impact from different settings, such as family, school and peers are addressed [70].

T1DM diagnosis is stressful for the children and their families. In the beginning, it is very hard for children to understand all the unexpected life changes; they often feel guilty or punished, they often express anger towards their parents, or/and fear of the unknown or of death. Parents, also experiencing fear and frustration, may use scare tactics and threats, which do not work, while, on the contrary, parents do constructively help their children by treating them as normal children and incorporating T1DM management as just an aspect of their daily routine [70,98,125]. Training children and parents to solve their problems efficiently, through strengthening their conflict-resolution skills, may facilitate the occurrence of supportive relationships, improve glycemic control, potentially reduce diabetes distress and finally improve quality of life [70,118,134,141].

The reorganization of family priorities and goals, regarding the development of supportive relationships and new daily routines, congruent with the “new normal” in family life, is crucial [65]. Family involvement, support and supervision, depends on the developmental stage and age of the youths, aiming to prevent a premature transfer of diabetes management to the child that would cause an uncontrollable deterioration in BG levels [70]. In addition, we must keep in mind the high rates of parental depression of children with T1DM, that usually occur when diabetes is firstly diagnosed [70,125,142] and that they are associated with higher risk of maladjustment and inefficient management, mainly in younger children [70,143].

#### 5.3.4. School Setting

Numerous studies provide data concerning the school-based intervention for youths with a chronic medical condition, since school setting is an ideal place, as children and adolescents, spend the most of their daily time there [96,97]. Managing insulin treatment in school setting, especially in younger children, requires the involvement of teachers, nurses and psychologists. In primary schools they should help youths to understand their condition’s seriousness and to manage their feelings of being different or of having a kind of disability [96,133,144]. The intervention at this early period of time, that the peer pressure is milder, is crucial as it will result a more positive attitude towards T1DM management later in adolescence [96,133,144]. Adolescence may disrupt the parental involvement in T1DM care. As youths increase their independence and gradually rely more and more to their peers, trying to “fit in”, they may hide their disease or even avoid T1DM management in school, compromising efficient glucose control [70,145]. In addition, *group interventions* conducted in school, significantly help children and adolescents to communicate and speak out their difficulties; and they should not only include diabetic patients, as a restriction that may stigmatize patients [96,133,144].

*Special education services* may help children with T1DM confront their learning difficulties. Schools may use protocols of response or/and intervention; the early intervention for school underachievement and the identification of children with SLD must be realized before the beginning of an education of special needs. The early intervention and identification of the cases aims to decrease the number of children that will automatically be evaluated for a special education without giving them the chance of an additional support, specifically a tiered academic support program in the frame of general education [27].

Additionally, as SLD has no “cure”, it may also be managed for a lifetime, so that youths, may enrich their learning skills in accordance to their differences [12,68]. Special education interventions, usually concern a multimodal education, incorporating multiple senses, helping students improve their skills in writing, reading, math [12,56,68] and through individualized instructions help them to compensate for their SLD [56,68]. Thus, reading skills may be improved through a structured and targeted approach, by strengthening decoding skills, phonological awareness, comprehension and achieving fluency [12,56,68]; improving writing focus on the writing and the composing process [12,56], by providing additional time on written assignments, or permitting the use of computer typing and by participating in classes with less number of students [12,56,68]. Academic interventions and expectations are also reviewed, from time to time, accordingly to the stage of development and the age of each child [12,56,68].

#### 5.3.5. Psychologist Intervention in the School Setting

School psychologists may provide vital support in both SLD and T1DM management into school setting, promoting success among students with chronic illness and facilitating accommodations in school and classroom, as well as the communication between, family, teachers and medical professionals [99].

### 5.4. Validation of the Therapeutic Frame/Interdisciplinary Approach/Intervention/Positive Outcomes

The positive results recorded during the monitoring of the child/follow up by the interdisciplinary team, relate to ensuring the well-being of the child and are reflected: (a) in child’s good mental and emotional health (b) in the good management of diabetes by the child and its family (c) in child’s good social functioning and its good school performance.

### 5.5. Limitations

Despite the fact that the review used a systematic approach-the strategy and criteria for the study inclusion mentioned above-, the method is subjective. The main limitation of the study is that only 12 articles were analyzed; furthermore, the selection bias via the inclusion criteria and the subjectively estimation of the chosen studies, may also augment the possibility of misleading in drawing conclusions.

### 5.6. Implication of Practice/Future Studies

The present research focusing on the enhancement of diabetes management and learning difficulties during childhood and adolescence through different settings, identified a minimally researched area. It could be seen as a bridge calling attention to the area and improve awareness and support within family and school setting, for children and adolescents with T1DM and SLD, aiming to develop a pilot study that may stimulate empirical research. The latter, to be used for evaluating the effectiveness of the multidisciplinary intervention, which will improve the patients’ physical and mental wellbeing, as first step to be towards this direction, the school attendance.

## 6. Conclusions

Although T1DM diagnosis and demanding treatment are a heavy burden for children and their families, T1DM may or may not be associated with a variety of academic and psychological outcomes. Despite the variability of the reviewed research design quality, it was clearly defined that the impact of T1DM is not uniform across educational and mental variables. Early interdisciplinary intervention is the key for children with SLD and chronic disease, such as T1DM. Supportive family and school, following the health professionals’ recommendations, is the most beneficial preventive approach for T1DM patients who experience educational problems, as indicated through both experience and research.

More extended research in this area is required, especially studies concerning the prevalence of T1DM and SLD; indicating the role of T1DM as a contributor or merely a correlate factor in educational and psychological outcomes; and focusing on the feasibility of insulin management programs for children and adolescent patients with the participation of both family and school context. This population group should be encouraged to participate in future research.

## Figures and Tables

**Table 1 brainsci-11-00004-t001:** Risk factors for Specific learning disorder (SLD).

Family history
SLD [13,21,22,23,27,29,30,31] Level of parental education [27,32] Special education services or educational supports [27,28,32] ADHD diagnosis [27,29,33] Not reading for pleasure [27,34] Genetic disorders [27,35]
Medical history
Prenatal and perinatal: premature labor or after risky pregnancies (diabetic gravidas), low/very low birth weight, complicated deliveries, hypoxia during labor and delivery, low Apgar score, neonatal jaundice, in utero substance exposure (e.g., alcohol, tobacco, radiation exposure, infections) [27,36,37,38,40] Other developmental (e.g., early speech-language delay [27,43] and mental health conditions (e.g., ADHD, disruptive behavior disorders, autism, anxiety disorders, and depression) [13,27] Neurocutaneous disorders (e.g., neurofibromatosis, Sturge- Weber syndrome, tuberous sclerosis complex) [27,35] Neurologic conditions or insults (e.g., seizure disorders, Tourette syndrome, history of central nervous system infection or irradiation or traumatic brain injury [27,29,35,42] Genetic disorders, syndromes or metabolic disorders, chromosomal disorders (e.g., fragile X syndrome, Turner syndrome, Klinefelter syndrome) [27,31,41,42] Medical conditions (e.g., recurrent otitis media, asthma) [27,29] Certain chronic medical conditions (e.g., T1DM, HIV infection) [13,27]
Socioeconomic status
low-income families/low socioeconomic status [13,24,25,26] cultural considerations [27,37,44] environmental disadvantage [27,37,44] poverty [27,34] under stimulating environments [13,24,25,26] neglect, abuse, domestic violence or unsafe home environment (e.g., parental substance abuse) [27,44] Adverse childhood experiences [27,45] Lack of adequate instruction [27,46]

Information from references [13,21,22,23,24,25,26,27,28,29,30,31,32,33,34,35,36,37,38,39,40,41,42,43,44,45,46].

**Table 2 brainsci-11-00004-t002:** Differential Diagnosis for Specific learning disorder (SLD).

Developmental delays (global and specific) [8,13,17,27,29,38,58,59,60,61] Genetic syndromes or metabolic disorders [8,13,17,27,31,41,42,58,59] Hearing impairment [8,13,17,27,58,59,62] Visual impairment [8,13,17,27,58,59,63] In utero substance exposure [27,38,40] Mild intellectual disability (formerly mental retardation) [8,13,17,27,36,58,59,60,61] Psychiatric conditions, ADHD or emotional disturbance (e.g., depression or anxiety) [8,13,17,58,59] Neurocutaneous disorders (e.g., neurofibromatosis, Sturge- Weber syndrome, tuberous sclerosis) [8,13,17,27,35,58,59] Neurologic conditions or insults (e.g., seizure disorders, Tourette syndrome, history of central nervous system infection or irradiation or traumatic brain injury [8,13,17,27,29,35,42,58,59] Seizure disorder (e.g., absence, partial, partial- complex) [27,29] Genetic causes [8,13,17,21,22,23,24,25,26] Parent/school expectations that are discordant with the student’s abilities and interests [8,13,17,58,59] Environmental factors (e.g., lack of opportunity, frequent school absences, poor teaching, and cultural factors, such as English as a second language) [8,13,17,58,59] lead poisoning, medication side effects, substance abuse [8,13,17,58,59,60,61] Sleep disorders [27,63]

Information from references [8,13,17,27,29,31,35,36,38,39,40,41,42,58,59,60,61,62,63,64].

**Table 3 brainsci-11-00004-t003:** Articles about impacts of T1DM on Learning, Cognitive, Psychological Function and Managing T1DM and SLD in family, school and medical setting.

Articles Reviewed	Authors	Method	Main Findings
Title/Year		Type/Population	
Effects of T1DM on learning and cognitive function in children and adolescents:			
1. Effects of Diabetes on Learning in Children. [92]. (2002)	McCarthy A.M., Lindgren S., Mengeling M.A., Tsalikian E., Engvall J.C. [92].	Three groups of children: (1) children with type 1 diabetes (*n*= 244), (2) a sibling control group (*n*= 110), and (3) an anonymous matched classmate control group (*n*= 209) [92].	For most children, type 1 diabetes was not associated with lower academic performance compared with either siblings or classmates, although increased behavioral concerns were reported by parents. The subtle cognitive deficits often documented in children with type 1 diabetes may not significantly limit the functional academic abilities of these children over time. Careful monitoring is needed to ensure that episodes of hypoglycemia associated with seizures are not adversely affecting learning [92].
2. Cognition and Type 1 Diabetes in Children and Adolescents [79]. (2016)	Cato M., Hershey T. [79].	A Meta-analysis/review. This article summarizes the existing literature examining the impact of glycemic extremes on cognitive function [79].	In children and adolescents with type 1 diabetes, exposure to glycemic extremes (severe hypoglycemia, chronic hyperglycemia, and diabetic ketoacidosis) overlaps with the time period of most active brain and cognitive development, leading to concerns that these children are at risk for cognitive side effects [79].
3. Impact of diabetes on cognitive function and brain structure [73]. (2016)	Moheet A., Mangia S., Seaquist E.R. [73].	Review of systematic reviews and Meta-analysis, longitudinal and cross-sectional studies, on the research that has been done over the last two decades to increase our understanding of how diabetes affects brain function and structure [73].	Both type 1 and type 2 diabetes are associated with mild to moderate decrements in cognitive function. They are significant differences in the underlying pathophysiology of cognitive impairment between type 1 and type 2 diabetes. T1DM is usually diagnosed at an early age and may have effects on brain development [73].
4. Cognitive functioning in young children with type 1 diabetes (T1D) [80]. (2014)	Cato M.A., Mauras N., Ambrosino J. et al. [80].	Neuropsychological evaluation of 216 children (healthy controls, *n* = 72; T1D, *n* = 144) ages 4–10 years [80].	Children with T1D were rated by parents as having more depressive and somatic symptoms. Learning, memory and processing speed were similar. Trends in the data supported that the degree of hyperglycemia was associated with Executive Functions, and to a lesser extent, Child IQ and Learning and Memory [80].
Psychological issues in children and adolescents with T1DM and psychiatric comorbidity:			
5. Depression and adherence to treatment in diabetic children and adolescents: a systematic review and meta-analysis of observational studies [93]. (2014)	KongkaewC., Jampachaisri K., Chaturongkul C.A., Scholfield C.N. [93].	Original article, Systematic review and meta-analysis of nineteen studies comprising 2935 juveniles [93].	This study showed moderate associations between depression and poor treatment adherence. Targeting behaviour and social environments, however, may ultimately provide more cost-effective health gains than targeting depressive symptoms [93].
6. Poor Metabolic Control in Children and Adolescents With Type 1 Diabetes and Psychiatric Comorbidity [94]. (2018)	Sildorf S.M., Breinegaard N., Lindkvist E.B., et al. [94].	Among 4725 children and adolescents with type 1 diabetes identified in both registers, 1035 were diagnosed with at least one psychiatric disorder [94].	High average HbA1c levels during the first 2 years predicted higher risk of psychiatric diagnoses. Patients with psychiatric comorbidity had higher HbA1c levels and an increased risk of hospitalization with diabetic ketoacidosis. Psychiatric comorbidity in children and adolescents with type 1 diabetes increases the risk of poor metabolic outcomes. Early focus on the disease burden might improve outcomes [94].
7. Psychosocial Status of Children With Diabetes in the First 2 Years After Diagnosis [95]. (1995)	Grey M., Cameron M.E., Lipman T.H., Thurber F.W. [95].	Children (*n* = 89 with IDDM, *n* = 53 without IDDM) ages 8–14 years were studied with the Children’s Depression Inventory, State-Trait Anxiety Inventory for Children, Child and Adolescent Adjustment Profile, Self-Perception Profile for Children, and a general health scale. Initial data were collected within 6 weeks of the diagnosis of IDDM and at 3, 6, 12, and 24 months thereafter [95].	After an initial period of adjustment, children with IDDM have equivalent psychosocial status to children without IDDM, but by 2 years after diagnosis, they have experienced twice the amount of depression and adjustment problems as their peers. Interventions should be aimed at this critical period between 1 and 2 years postdiagnosis [95].
Managing T1DM and SLD. Studies investigating the family, school and medical setting for support for children with Type1 Diabetes:			
8. Type 1 Diabetes in Children and Adolescents: A Position Statement by the American Diabetes Association [70]. (2018)	Chiang J.L., Maahs D.M., Garvey K.C., Hood K.K., Laffel L.M., Weinzimer S.A., Wolfsdorf J.I., Schat D. [70].	Position Statement reviewed and approved by the American Diabetes Association Professional Practice Committee and ratified by the American Diabetes Association Board of Directors in 2018 [70].	This Position Statement was developed under the 2017 criteria (7) and provides recommendations for current standards of care for youth (children and adolescents) with type 1 diabetes. The majority of pediatric recommendations are not based on large, randomized clinical trials but rely on supportive evidence from cohort/registry studies or expert consensus/clinical experience [70].
9. The effect of group therapy on Diabetes specific Knowledge [96]. (2015)	Hankins M.A. [96]	Unpublished theses, dissertations and capstones. Paper 947. Marshall University. A school-based study evaluated the effectiveness of a psychoeducational group therapy intervention on Diabetes Specific Knowledge (DSK) [96].	Children participating in the psychoeducational group intervention increased their overall Diabetes Specific Knowledge [96].
10. School-based tertiary and targeted interventions for students with chronic medical conditions: Examples from type 1 diabetes mellitus and epilepsy [97]. (2008)	Wodrich D.L., Cunningham M.M. [97].	Using epilepsy and type 1 diabetes mellitus as examples of two conditions associated with a risk of school problems, this article outlines roles for school psychologists and provides specific guidance about how they can promote success among all students with chronic illnesses [97].	As health service providers, school psychologists understand both the educational process and the ways in which childhood illnesses can impact it. This article argues that school psychologists’ breadth of knowledge enables consultation with teachers about health-related classroom accommodations and communication between medical professionals and educators [97].
11. Psychological issues in the care of children and adolescents with type 1 diabetes [98]. (2005)	Frank M.R. [98]	Review article that highlights some of the psychological issues in children and adolescents with type 1 diabetes and provides health professionals with some strategies for addressing them. [98].	This research shows that the best preventive approach to the psychological difficulties seen in children and adolescents with diabetes is a strong, supportive family who is able to gain strength and direction from a team of professionals sensitive to the psychological issues associated with diabetes and who act on them appropriately [98].
12. The importance of psychological counseling in reducing symptoms of depression and increasing self-esteem of children with diabetes [99]. (2012)	Dronjaka D., Kesic A., Cvetkovic M. [99].	This study evaluated 19 children (from 10 to 16 years) with diabetes, using Children’s Depression Inventory and Self-Esteem Inventory. During the two years (once a week), they attended psychological group workshops and individual treatments [99].	This study determines the importance of psychological support in reducing serious physical, mental, and emotional challenges that diabetic youth confront, as they have greater rates of depression and lower self-esteem. The control results showed a significant reduction in the degree of depression (32%) and higher score in self-esteem inventory (38%), especially at subscales school and social. This research has demonstrated the efficacy of psychosocial therapies children with diabetes and necessity of developing psychosocial intervention programs [99].

**Table 4 brainsci-11-00004-t004:** Therapy Framework for children and adolescents with T1DM and SLD.

Early intervention and follow up for T1DM adherence:
In-depth education and behavioral interventions are mostly provided the period following diagnosis, to prevent the development of negative habits [98,137]. Ensure that the family has the opportunity to learn about T1DM and get prepared for the psychological impact of its diagnosis [98]. Ensure not to overwhelm the family and in the first few days, to provide only key ‘survival’ information and allow taking time for grief [98]. Recommend early referral to the child-adolescent psychiatrist for assessment and support [98] and provide appropriate referrals to mental health professionals [70]. Assess psychosocial issues and family stresses that could impact diabetes management. A thorough, age- appropriate psychosocial evaluation and review of the medical regimen will suggest targets for modification to facilitate self- management and well-being [70]. Assess social adjustment (peer relationships) and school performance to determine whether further evaluation is needed [70]. Assess generic and diabetes-related distress, generally starting at 7–8 years of age [70]. Evaluate disordered or disrupted eating behaviors using validated screening measures when hyperglycemia and/or weight loss are unexplained based on self-reported behaviors related to medication dosing, meal plan, and physical activity. In addition, a review of the medical regimen is recommended to identify potential treatment-related effects on hunger/caloric intake [70]. Consider and evaluate the resistance of patients to accepting support from clinicians, family, and friends, as a possibility of a more serious psychological issue [70], in case of persistence for a long period of time and despite the occurrence of serious difficulties in self-management.
SLD prevention of school failure-evaluation of children and adolescents with a potential SLD:
Assess for several early risk factors for SLD A medical examination should involve a detailed history, including a review of medical risk factors (e.g., premature birth, very low birth weight; according to Table 1) [27,28,29,30,31,32,33,34,35,36,37,38,39,40,41,42,43,44,45,46] and current comorbidities (e.g., speech-language delay or disorder; that are associated with SLD) [27,29,30,37,39,43,60,62,63,65,66,67], and hearing and vision assessment if not recently assessed. [27,62,64] Relevant physical examination findings (e.g., head circumference, birthmarks, facial morphology). Consider key differential items that may contribute to academic underachievement [56] Consider where treatment within the medical field (e.g., cognitive behavior therapy, medication) will help with educational needs [27,29,31,35]. Symptom identification and screening resources SLD [27,36,38,39,40,41,42,56,60,61,62,63,64]. Children with more complicated medical disorders (e.g., T1DM) or concerning risk factors (e.g., family history) need to be treated by pediatric subspecialists. This would include children with insulin who have academic underachievement when glucose levels are low and require ongoing subspecialty treatment through insulin or pediatric endocrinology [27].

Information from references [27,28,29,30,31,32,33,34,35,36,37,38,39,40,41,42,43,44,45,46,56,60,61,62,63,64,65,66,67,70,98,137].

**Table 5 brainsci-11-00004-t005:** Age- based care-Typical development and diabetes demands and priorities across childhood and adolescence.

Developmental Level (Corresponding Ages)	Typical Developmental Tasks	T1DM Management Priorities (and Person/Care Giver Who Is Responsible)	Family and School Considerations Due to Presence of T1DM
infancy and start of toddlerhood (0–2 years)	Attachment and development of trusting bond with caregivers Physical development and reaching milestones of first words and walking [70]	Reduction of wide fluctuations in glucose levels (caregiver) Prevention of hypoglycemia (caregiver) [70]	Vigilance in identifying child symptoms of hypo- and hyperglycemia Coping with stress associated with management and additional responsibilities [70]
end of toddlerhood through early childhood (2–6 years)	Often begin formal schooling- preschool to elementary school Separating from caregivers for activities Physical growth with interests in exploring new challenges and activities [70]. Rapid period of physical and neurological development with frequent, inconsistent bursts of physical activity level [131] Language Skills: Difficulty verbalizing thoughts and feelings [131] Cognitive development: Concrete thinking [131] Social Development: Children may be worried about being away from parents but may also become interested in spending time with others at friend’s house [131] Emotional and behavioral development: Increased resistance to/fear of doctors, the hospital, and needles [131]	Reduction of wide fluctuations in glucose levels (caregiver, school personnel) Prevention of hypoglycemia (caregivers, school personnel) [70]; Trusting others to help with diabetes management (teachers, friends’ parents, family members), including how to recognize signs of high or low BG levels and treatment [70,131] Parental worry about hypoglycemia may impact daily management [131] May be difficult to get an active child to remain still for injections, and blood glucose (BG) monitoring; Increased resistance to, or anger about, injections and BG checks; Children want to make more of their own choices (e.g., eating, clothing, where to do BG or place pump site) [131] It can be hard for parents to distinguish low or high BG levels from a “normal” tantrum or bad mood [131]	Continued vigilance in identifying child symptoms Communicating with school and planning for monitoring when not with child; coping with stress [70] Necessitates frequent monitoring and adjustments to insulin and nutrition needs during growth spurts [131]; Close monitoring of food intake and adjustments for variable appetites [70] Requires increased monitoring of activity and related glycemic variability [131] It might be hard for young children to communicate symptoms of high or low BG levels, worries, or questions about T1DM care; and they may not understand why daily T1DM management tasks are required, such as why insulin is needed, need for BG monitoring, or why they may not always eat the same types or amounts of food that their friends eat [131] Often have specific fears; Temper outbursts are common; Sometimes want to do things “their” way or by themselves; Learn ways to manage their feelings as they grow and have new experiences [131]
late childhood (7–11 years)	Developing skills in physical, social, and academic areas Gaining more autonomy from primary caregivers, yet still very reliant on caregiver supervision Often engaging in team activities that promote sharing and understanding views of others- empathy growth [70]	Sharing in the identification of symptoms of hypo- and hyperglycemia (child and caregiver) Treating hypoglycemia and carrying supplies (child with supervision from adults) Developing sense of problem solving and flexibility with regimen if plans or activities change (child with guidance/modeling from caregiver) [70]	Teaching child symptoms of hyperglycemia and hypoglycemia and basics of diabetes management and treatment Praising conduct of management tasks and modeling problem solving when new diabetes problems arise Coping with stress and new challenges of complex schedules and eating patterns Helping teach child to disclose to others about diabetes [70]
Early adolescence (12–15 years)	Managing physical and emotional changes Attempts at “fitting in” with peer groups; peers becoming larger influence on behavior Developing stronger sense of self and identity Teens desire less guidance and supervision from caregivers, yet still needing it; Disclose to others about diabetes for safety [70]	More decision making about diabetes management and regimen changes for teens Teens’ expectation to monitor and be vigilant about glucose excursions when away from primary caregivers Parents take to respect the privacy of teens /young adults, especially regarding behaviors that are considered taboo or risky [70]	Developing new forms of monitoring and communicating about diabetes Coping with common increase in conflict about diabetes management Supervising enough but attempting to support growing autonomy in teen [70]
Late adolescence (16–19 years)	Expansion of networks and activities (driving) Increased thinking and worries about what is next Expectation to make decisions based on interests and opportunities [70] Risk behaviors, such as alcohol, smoking, drug use and unprotected sexual intercourse [138] Increased risk of psychiatric disorders (primarily depression and eating disorders) [138]	Increasing autonomy for many management tasks (teen) Diminishing seeking of guidance and supervision from caregivers (teens) Discussions about transition to different diabetes care providers (teens, care team, and caregivers) [70] Screening for depression and experimentation with risky behaviors [138]	Balancing need for supervision and guidance with less face-to-face time with teen and more teen autonomy Modeling positive decision making about diabetes and life choices Creating liaison with different diabetes care providers for transition to adulthood [70] Counseling about smoking avoidance, use of contraception should be reviewed and encouraged [138]

Information from references [70,131,138].

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
