# Peer review of "Priorities in the Interdisciplinary Approach of Specific Learning Disorders (SLD) in Children with Type I Diabetes Mellitus (T1DM). From Theory to Practice"

_brainsci, 2020, doi:10.3390/brainsci11010004_

Round 1

Reviewer 1 Report

I must admit that prior to reading the paper, I was inclined to reject it solely because it purported to be a review of just 12 studies.   Upon reading the manuscript, my opinion has changed.   The authors have identified a minimally researched area, and by calling attention to the area, they may stimulate empirical research.  

I think the highlight of the article was the variability of reported research.  It appears that a diagnosis of T1DM may or may not be associated with a variety of educational and psychological outcomes.   Authors note the variability of research design quality, but make it clear that the impact of T1DM is not uniform across educational and psychological variables.   This has considerable value, largely in that it invites more research in the area.   Is T1DM a causal contributor or merely a correlate?  I like that the manuscript invites future research.

I would suggest that whenever reference is made to the impact of T1DM, perhaps the authors would insert the word "potential." 

Reviewer 2 Report

The manuscript "Priorities in the interdisciplinary approach of specific learning disorders in children with type I diabetes mellitus. From theory to practice. A narrative review of the literature" propose an interesting and pertinent review. It is well structured and well written. I do not have any comments to suggest. I am favorable to the publication.